# Analysis of Passive RFID Applicability in a Retail Store: What Can We Expect?

**DOI:** 10.3390/s20072038

**Published:** 2020-04-05

**Authors:** Maja Škiljo, Petar Šolić, Zoran Blažević, Toni Perković

**Affiliations:** Mechanical Engineering and Naval Architecture in Split, Faculty of Electrical Engineering, University of Split, 21000 Split, Croatia; zblaz@fesb.hr (Z.B.); toperkov@fesb.hr (T.P.)

**Keywords:** Internet of Things, Radio Frequency IDentification (RFID), passive tag, tag antenna, multipath propagation, retail store, Software Defined Radio

## Abstract

The Internet of Things (IoT) has a lot to offer and contribute to the retail industry, from the innovations in retail store experience to the increased efficiency in the store management and supply chain optimization. On its way to real-world applications, Radio Frequency IDentification (RFID) became the main enabler for the final IoT deployment. However, to improve the technology performance even further, it is important to overcome the fundamental limitations of its physical layer and, consequently, to better understand how to use the technology in an optimal way. The analysis provided in this paper employs the simulation/measurement study on RFID technology advancement and the influence of radio propagation in a realistic model of the retail environment. The results are provided for different types of the retail layouts and materials that influence tag responsiveness.

## 1. Introduction

The Internet of Things (IoT) is slowly changing everyday life and industries across the board. It merges physical and digital worlds by connecting all kinds of devices and enabling them to communicate in real time using digital intelligence. Today, there are several main real-world IoT applications: healthcare, smart home and cities, automotive applications, agriculture, energy, and the industrial and retail sector. Retail IoT application offers consumers a bridge between online and in-store shopping. Going towards IoT in a retail store, it is necessary to observe the problem of store setup from the electromagnetic (EM) point of view. The fundamental limitations of IoT or RFID technology actually arise from EM constraints, antenna radiation properties, and radio frequency (RF) propagation phenomena.

It becomes apparent that the most affordable IoT technology for identification, detection, and tracking is RFID. Passive Gen2 RFID systems [1] are more attractive to IoT applications than the active ones due to their low cost, small size, simple design, flexible and embeddable tag structure, longer duration, wider range of applications, and lower maintenance [2].

Retail store application for worldwide use practically implies low-cost solutions, i.e., common passive RFID tags. There are other types of tags, like on-metal and on-product tags, but this requires specific design, limited application and higher cost [3,4]. Today, the retail store is still organized according to people, i.e., customer needs, which causes the arrangement of products and shelves suitable for customer’s choice of food (all together in one place, i.e., all sweets on the same shelves) and their pattern or habits for shopping with shopping baskets and trolleys. The characterization of EM phenomena in a retail store may influence and help the reorganization of IoT store setup: the position of reader antenna(s), the layout of the shelves according to better signal coverage and choice of their material, arrangement of product sections according to their material, placing premium products in high traffic areas, etc.

To cope with the issues of optimal RFID deployment, a wide variety of contributions have been published in the past. The problems were studied from different perspectives ranging from software solutions that organize timely tag delivery, to hardware proposals that improve tag responsiveness. For example, one studied the traceability through the food production chain and used equipment to read the tagged items in the supply chain [5,6], where other possible related software architecture was presented in [7]. Later on, different EM issues related to the materials that tags are attached to were preliminarily analyzed for purposes of understanding the influence of material [8], or even approaches to design new tags that should have improved performance if attached to liquids [9]. Deeper evaluation of performance starting from simulations and a radio-link budget point of view was presented in [10], while first-presented empirical evaluation of the commercial reader was given in [11]. To characterize the performance, different tools (such as sniffer) for the evaluation of a commercial reader were presented in [12]. Also, the influence of different protocol parameters was evaluated in [13]. A newer approach [14] provided expected, simulation-based performance in an interference scenario. However, what is missing in the literature is a useful empirical study on how well new RFID systems behave. A similar approach was done in the office environment [12], with an older generation of the technology. The aim of this paper is to understand how much the technology has advanced in terms of reading distances and tag read reliability, and what can be expected if placed in a realistic scenario, such as a standard retail store.

Common passive RFID tags are powered up by the incident RF power, which is backscattered and modulated by the tag antenna and chip. Even when this incident RF power is constant, and the RFID system is stationary and positioned in the environment with no obstacles whatsoever, the system performance depends on power level, frequency of operation, and orientation of the tag with respect to the reader. This is due to chip and antenna characteristics. Here we focus on the variation of power impinging on the tag and a mixture of material properties surrounding the tag. In a retail store there is a wide variety of propagation phenomena due to a complex environment full of furniture, shelves, refrigerators, and products made of materials with different electrical properties.

To characterize the retail store as an IoT environment, a lot of measurements and simulations with the help of fundamental theory are needed. It is virtually impossible to calculate and take into account each product’s size, material, and propagation effects for every retail store. Therefore, the empirical study is performed to characterize the application of RFID in a retail store from an EM point of view. The contribution of this study is the evaluation of tag responsiveness in a real retail store by EM characterization of a multipath channel and tag antenna radiation properties. RFID signal propagation phenomena in a store simulation model equipped with standard retail furniture (shelves, desks, refrigerators, etc.) are investigated for different scenarios, from empty shelves to shelves with products of different electrical properties. In addition to simulations, measurements performed in a proof-of-concept manner are used to estimate the technology advancement, i.e., tag performance for different generations of chips. The tag attached to different retail store products is analyzed, and the key performance drawbacks are pointed out. In a real retail store, the EM properties are evaluated by the measured number of tag responses in line-of-sight (LOS) and non-LOS area, and with the tag attached to retail products.

The first part of the paper, Section 2 deals with the fundamental aspects of the RFID system and points out the main EM effects of retail products on tag antenna radiation properties. Section 3 focuses on a retail store simulation model, different products’ materials, different receiver setups, and visualization of EM propagation phenomena. Here, different generations of chips are compared by their sensitivity level to how it affects the reader coverage in the store. The measurement results are given at the end of this section, and tag responsiveness is evaluated by the EM characterized phenomena.

## 2. Electromagnetic Characterization of Tag Antenna Attached to a Retail Product

First, we analyze the fundamental aspects of a monostatic RFID system in free space. The minimum sensitivity level of modulated backscattered tag signal power $P_{rx,sens}$ at the receiver can be calculated for a maximum tag range $r_{max}$ according to the standard radar equation [15]:(1)$$P{}_{rx,sens}=P_{tx}G_{reader}^{2}\left(\theta ,\phi \right)p\frac{\lambda^{2}}{{\left(4\pi \right)}^{3}r_{max}^{2}}\Delta \sigma$$
where λ is the wavelength, $P_{tx}$ the reader’s transmitted power, $G_{reader}\left(\theta ,\phi \right)$ the gain of reader’s transmitting and receiving antenna in tag direction, θ elevation angle, ϕ azimuth, and *p* the polarization efficiency which accounts for polarization mismatch between tag and reader antenna. In our case, the polarization efficiency is 0.5 (which decreases $r_{max}$ by $\frac{1}{\sqrt{2}}$) because the reader antenna is circularly polarized and tag antenna linearly polarized [16]. Differential radar cross section (RCS) Δσ of a tag depends on wavelength, impedance match between the antenna and chip, and tag antenna gain $G_{tag}\left(\theta ,\phi \right)$ in the direction towards reader, as [15]:(2)$$\Delta \sigma =\frac{\lambda^{2}}{4\pi }G_{tag}^{2}\left(\theta ,\phi \right)K$$
where *K* is modulation loss caused by specific modulation details α (necessary for calculating the time average signal power and here taken to be α = 1 [17]) and reflection coefficients $\Gamma_{low,high}$ between the tag antenna and two states of chip impedance, $Z_{a}$ and $Z_{low,high}$ [15], calculated by:(3)$$K=\alpha {\left|\Gamma_{high}-\Gamma_{low}\right|}^{2}$$
(4)$$\Gamma_{low,high}=\frac{Z_{low,high}-Z_{a}^{*}}{Z_{low,high}+Z_{a}}$$

Passive RFID system tags modulate backscattering power by switching the chip impedance between two modulation states. The degree of backscatterring depends on this modulation efficiency; the higher the better [15]. The gain of tag antenna is a key performance parameter obtained from antenna directivity *D* and radiation efficiency $e_{r}$ that accounts for dielectric and conduction losses of the antenna, $G\left(\theta ,\phi \right)=D\left(\theta ,\phi \right)e_{r}$ [18]. When a tag is attached to a retail product of certain electrical properties different from air, the antenna’s impedance, radiation efficiency, and directivity are changed, causing lower backscatter or backscatter in a direction different from the preferred one.

In computational electromagnetics software FEKO (from the German acronym ‘FEldberechnung für Körper mit beliebiger Oberfläche’, in translation as ‘Field Computation Involving Bodies of Arbitrary Shape’) we simulated the ALN9640 tag (Alien Technology, Morgan Hill, USA), whose structure in free space is shown in Figure 1. Tag antenna consists of a meander line dipole and T-match element that acts as an impedance transformer. The copper tag structure is modeled using Method of Moments (MoM) with triangle mesh elements. As various food and packaging have different electrical properties, from metallic cans, liquids, to plastics [19], we simulated this tag attached on a cylinder presenting a retail product shown in Figure 2 to investigate the material effect. For the purpose of the analysis in this paper, the retail products’ materials are generalized and represented with three basic materials: metal (representing all cans and items with metallic packaging), water (representing all liquids) and plastic (such as a low-loss dielectric for all non-metallic and non-liquid retail products). Plastic properties $\epsilon_{r}=3.5$ and σ=0.002 S/m are taken as a general low-loss dielectric to cover the whole range of retail products like flour, sugar, plastic products, etc. [19,20]. Water properties are $\epsilon_{r}=81$ and σ=0.22 S/m (from Wireless InSite material database [21]; see Table 2).

The 3D radiation patterns of these tags are simulated at a US RFID frequency of 920 MHz. Figure 2 and Table 1 show three effects when the tag is attached to the product made of plastic, water and metal:The first effect, in Figure 2 demonstrates the deformation of the radiation pattern, i.e., antenna directivity in all cases. A metal product in the form of a cylinder is reacting like a reflector or practically additional radiating part of tag antenna, causing the radiation in the *z*-axis direction. However, in the case of dielectric (plastic and water), all radiation goes in a material direction. The higher the dielectric constant, the larger the deformation of the radiation pattern. This causes a change of tag antenna directivity D(θ,ϕ) in the direction of the preferred one (towards reader antenna or omnidirectional as in free space).For the second effect, the maximum gain of the tag antenna is decreased when attached to the examined materials because of their conduction and dielectric losses (also shown in Figure 2). Generally, as the water content gets higher in a product material, the greater the losses become, e.g., the increasing moisture content in corn flour causes this effect. The gain is decreased, because these losses lower the radiation efficiency of tag antenna. As shown in Table 1, radiation efficiency is significantly decreased when the tag is attached to a product, especially metal.The third important effect, shown in Table 1, is the change of antenna impedance when the tag is attached to a product. This influences the proper match with chip’s impedance (reflection coefficients in Equation (Equation 2)) and consequently lowers the power backscattered by the tag due to a lower level of absorbed power (from Equations (Equation 1) and (Equation 2) one can see that tag range $r_{max}$ is proportional to the matching factor (Equation 4)). The results for antenna impedance in Table 1 for all examined materials show a great degree of lowering both real and imaginary antenna impedance when compared to a tag antenna without product.This means that for an assumed Higgs 3 chip’s impedance of low and high state [22] at 920 MHz, the matching factor (Equation 4) can vary from 2.96 on water to 0.00067 on metal cylinder. It is important to note that the chip’s impedance in a modulated state is hard to measure and it depends on frequency and power impinging on the chip (if translated to distance this becomes an equivalent of the tag range).

The influence of product materials on tag antenna performance is generally bigger when the product is larger in size. On the contrary, it can be decreased when the tag antenna is not closely attached to the product or just attached partially. These effects are examined, and the results of tag antenna properties attached to a plastic cylinder in three different scenarios are shown in Table 2. The directivity results are depicted in Figure 3. The influence of positions and the size of product on tag performance can be summarized as follows:The first scenario increases the separation between the tag antenna and the product by 3 mm and 6 mm. This causes the antenna impedance and radiation efficiency to approach the free space values. However, if we look at the gain and directivity results in Table 2 and Figure 3 (left), the deterioration of radiation pattern is still present. The gain and directivity can even be higher in these cases than in free space (this is due to a larger radiating structure), but the direction is not the preferred one.The second scenario is a tag on a larger dielectric product. Again, the product is represented by a cylinder of the same position and orientation, but twice the size. For example, the effect is not equal when a tag is attached to a small bottle of water, or on a larger water canister. The influence of a product is more emphasized in that case, which can be seen from the all results given in Figure 3 (center) and Table 2.The third scenario is tag antenna partially attached to a product. Figure 3 (right) shows the radiation pattern for the tag antenna with the major part attached to the product. This also causes deterioration of a radiation pattern but in directions other than the previous ones (due to the unsymmetrical radiating structure).

## 3. Influence of Retail Store Products and Propagation Phenomena on a Tag Sensitivity

The influence of the environment has been analyzed by the variation of the signal impinging on the tag antenna. Even for a standalone tag (not attached to any product) at the range lower than $r_{max}$ there is a possibility of low or no readings due to the radio channel, i.e., obstacles (depending on their size, material and position) in the vicinity of reader antennas and tags.

In radio channel modeling, the wide-band channel is represented as a sum of multiple copies of an original transmitted signal, i.e., multipath components. This is because different replicas of the transmitted signal travel over electrically different path lengths. In indoor environments, each signal path suffers from various phenomena: reflections, diffractions, transmissions, and scattering. Thus, at the receiver, we get the sum of multipath components, with amplitude attenuated mostly due to lossy materials interacting with the signal or radio shadow, and different propagation delays due to different path lengths. In an idealized channel, when a Dirac impulse is applied at the input x(t)=δ(t) the complex channel impulse response h(t) (in equation where asterisk denotes the convolution operator), identical to the output signal y(t), is given by:(5)$$y\left(t\right)=\delta \left(t\right)*h\left(t\right)=h\left(t\right)\overset{}{\to }h\left(t\right)=\sum A_{n}e^{j\phi_{n}}\delta \left(t-\tau_{n}\right)$$
where $A_{n}$ is the path gain, $\phi_{n}$ the random phase (uniformly distributed), and $\tau_{n}$ the propagation time delay of each multipath component. Together they represent the specific pair ($A_{n}$,$\tau_{n}$) which describes and indicates each copy of the transmitted signal in the wide-band propagation channel [23].

### 3.1. Simulation Model and Setup

For the purpose of obtaining the realistic simulation model of a retail store, a radio propagation software Wireless InSite is used [21]. Ray-tracing models and high-fidelity EM solvers are used for the analysis of wireless communication systems and radio propagation channels. It gives efficient predictions of EM propagation and, also important, the visualization of the EM wave propagation for better understanding of the scattering phenomena in complex environments.

A real retail store was built in Wireless InSite according to the given floor plan with dimensions of 8.6 × 6.2 × 3.1 m, concrete floor and ceiling, metallic shelves, refrigerators, wooden baskets and beverage packs as shown in Figure 4 for different views. Walls depicted with light gray color are made of concrete; white color represents glass material; dark gray is metal; brown is wood, and blue is for water. All materials’ electrical properties in the model, except for plastic, are listed in Table 3. Their parameters are taken from Wireless InSite database of material properties given or calculated according to International Telecommunication Union (ITU) references [21]. The calculation is done with the X3D propagation model with five reflections, two transmissions, one diffraction, and ray spacing of 0.25 m for each receiver considered. The transmitter antenna is a directional circularly polarized antenna, with gain of 5 dBi supplied with power of 30 dBm at f=866 MHz, whereas the receiver tag antenna is modeled as an omnidirectional horizontally polarized antenna with 2 dBi gain (as in FEKO simulations in free space, see Figure 1). The transmitter was placed at one place for simulations and measurements, whereas three different receiver setups were used to examine the impact of receiver height, random position in the store by taking some route around the shelves. Also, various scenarios of products’ materials, position and setup are investigated: store with empty shelves and refrigerators; store with shelves and refrigerators filled with products of water (water is representing all liquids, as in Section 2), metal and plastic; mixed products positioned in the simulation model as in a real store and the last one, mixed products scenario in which the receiver antenna has significantly lower gain (−12 dBi) in order to represent the case when the tag is attached to a bottle of water.

#### Propagation Paths in LOS Area

In this subsection the visualization of the propagation paths is provided in the form of 2D, 3D and graph, which is important for further understanding of the EM phenomena in a retail store. For each receiver position 25 rays are calculated, i.e., their propagation parameters by (Equation 5), which ensures the convergence of the results within a reasonable calculation time. The multipath components with the longest path arrive to the receiver with larger delays, and the ones that suffer multiple reflections and transmissions show larger attenuation. Figure 5 shows the channel impulse response where the path gain $A_{n}$ vs propagation delay $\tau_{n}$ is given for receiver position 11 (in receiver grid 1) with 2D and 3D view of propagation paths. It is shown that the received power at each receiver position depends on the multipath channel, i.e., on the scattering from the objects in the vicinity of reader and tag antenna. Even in LOS between the reader and tag, there is a possibility that tag is not read due to multipath reception.

### 3.2. Chip’S Generation Comparison and Effect of Retail Products on the Shelves

So far, we have investigated tag antenna performance, but it is also well known that the chip’s response depends on frequency and power impinging on the chip [24]. Also, there is a tag activation power threshold needed for the tag to respond at each frequency. The measurement platform for UHF RFID tags characterization [24] was used to determine the $P_{tag,sens}$ of three tags from one company (Alien Technology, Morgan Hill, USA) with similar layout but with different chips (Higgs 2: ALN-9540, Higgs 3: ALN-9640 and Higgs 4: ALN-9740) throughout the RFID frequency band.

The results, given in Figure 6, show that generally the new generation of chip shows better (lower) $P_{tag,sens}$ in comparison with the previous ones. Also, it can be noticed that in European UHF RFID frequency band (865–868 MHz) the difference between the Higgs 3 and 4 in relation to the Higgs 2 chip is larger than for US UHF RFID frequency band (902–928 MHz). Therefore, in this band we can expect bigger differences, i.e., better improvement in tag range and readings, when using a new generation of chip instead of the old one.

In our work, the results are obtained at *f* = 866 MHz whereas Higgs 2, Higgs 3, and Higgs 4 achieve the sensitivity level $P_{tag,sens}$ of −14.3 dBm, −18.4 dBm, and −18.5 dBm, respectively. As the sensitivity for Higgs 3 and Higgs 4 are very close, in further text, the results and discussion are given only for Higgs 2 and Higgs 4 comparison, but the conclusions are also valid and can be applied to Higgs 3 and the rest of the RFID frequency band.

#### 3.2.1. Receiver Positions in Front of the Shelves

The measured sensitivity levels are used in the simulation store model to depict reader’s coverage of signal level necessary to power up the tag (red is for the power above the tag’s $P_{tag,sens}$ and blue is for below, Figure 7). In this setup, receivers are positioned in vertical grids in front of the shelves to examine how can $P_{tag}$ varies with height. Figure 7 depicts the coverage of the reader signal for Higgs 2 and Higgs 4 where the shelves are filled with products of mixed properties as in the real store. These results show that due to multipath propagation, different power levels are impinging on the tags depending on their position on the shelves. Due to 4.2 dB difference between the $P_{tag,sens}$ of Higgs 2 and Higgs 4, there is an implication that at some shelves where the power level is low and near the threshold, Higgs 4 can be powered up, but Higgs 2 cannot.

#### 3.2.2. Random Receiver Positions Around the Retail Store

In the second receiver setup, different chips’ generations Higgs 2 and 4 are investigated for the transmitter position shown in Figure 4 where the receivers are positioned in the form of grids around the store (Figure 8, Figure 9 and Figure 10) at the same height of 1.5 m, i.e., the height at which the person is carrying the product. Three scenarios are depicted: empty shelves with a standalone tag, shelves with products of mixed properties and standalone tag, and again shelves with mixed properties but with a tag attached on a product (neglecting impedance mismatch and directivity deterioration, just taking into account the radiation efficiency and consequently antenna gain drop). The gain of the receiver antenna attached to water product is −12 dBi and directivity pattern remained omnidirectional. The last scenario is taken as the most realistic one where just one of the material effects (see Section 2) is taken into account to isolate the EM effects on tag performance. It can be expected that power distribution in the store would be changed if the directivity deterioration is taken into account because the tag signal in that case is received from random, generally non-preferred directions of propagation paths. This is a part of the authors future work on the subject.

The results show the power distribution in the store where multipath propagation causes drops of the signal even in LOS area. In all cases, Higgs 4 shows better coverage than Higgs 2, due to 4.2 dB better sensitivity level. When comparing the investigated scenarios, by placing the products of mixed properties on empty shelves and refrigerators, the multipath channel changes and the spots where tags do not receive enough power can happen even in places close to the reader and in the LOS zone. In the most realistic scenario, where the tag is attached to a product and its material effect is taken into account, the coverage of necessary tag activation power threshold is very limited, especially in the case of Higgs 2. The lower tag antenna gain decreases the tag range $r_{max}$, and in this case, each decibel of the received power can be important for powering up the tag. The power can probably be gained by a different setup of reader antenna(s), arrangement of the store, orientation of tags, non- metallic shelves, etc.

Besides power distribution in the store, we investigated the cumulative distribution function (CDF) of $P_{tag}$ at each receiver position for all examined scenarios of a 2D receiver grid, Figure 11 and Table 4. The power level in decibels exhibits an excellent fit to a Gaussian distribution typical for indoor multipath propagation, as shown in Figure 11 just for the scenario with mixed products on the shelves with lower tag antenna gain. The probability that Higgs 2 will not be read (the power impinging on the chip is lower than $P_{tag,sens}$ = −14.3 dBm) is generally around 50% and higher than Higgs 4 results by cca 10–20%. Placing the metal products on empty shelves improves the multipath gain, i.e., the mean of $P_{tag}$ is increased and its STD (Standard Deviation) decreased, whereas the plastics on the shelves bring the lower mean and higher STD. All scenarios give practically similar results of mean and STD of $P_{tag}$, except for the scenario with mixed products and lower tag antenna gain. Therefore, for this setup of reader antenna and store arrangement, a mean of $P_{tag}$ is the lowest, the STD the highest and there is a possibility of only 19% that Higgs 4 will be read, and 13% for Higgs 2. This means that lowering the gain of tag antenna (by attaching the tag on some product) can influence the probability of being read significantly. In our case, by introducing the lower gain of tag antenna (see last two rows in Table 4), for Higgs 2 the probability of not being read increased by 38% and for Higgs 4 by 49%.

In both receiver scenarios, vertical and 2D receiver grids, it is shown that the improvement of a chip’s sensitivity level influences the reader coverage whereas the product influence is more emphasized when attached to a tag antenna than in the case when it is just placed on the shelves.

### 3.3. Retail Store Results

#### 3.3.1. Simulation Model of a Measurement Scenario

As the retail store model with different receiver setups was analyzed in previous sections, first the receiver trajectory scenario was simulated as performed in measurements for comparison and easier visualization of the measurement results. Figure 12 depicts the trajectory of a Higgs 4 tag (attached to cardboard and water product as in measurements) around the metallic shelves with products of mixed properties and arrangement (as in the real store during the measurements). Besides the power distribution along the trajectory, the results are also given in a form of 2D plot of $P_{tag}$ vs receiver number, Figure 13, to highlight the variation of a signal level along the trajectory which covers LOS and non-LOS area (behind the shelves). Again, the receiver was an omnidirectional antenna in free space with gain $G_{tag}$ = 2 dBi (representing tag antenna on a cardboard) and omnidirectional antenna of lower gain, $G_{tag}$ = −12 dBi (representing tag antenna on a water product). From these results, we can see that due to multipath fading, $P_{tag}$ drops significantly in a non-LOS area and even in LOS at certain receiver positions (receiver positions 3, 6 and 35, see Figure 12 and Figure 13) when the tag is attached to a water product. The variation of $P_{tag}$ along the trajectory can even reach 20 dB from one receiver position to the next. On the contrary, when the tag is in free space, this variation is very small. The comparison of $P_{tag}$ along the trajectory in free space and in a retail store shows the nature of multipath; in certain positions the received power is higher than in free space, and in some, much lower due to constructive and destructive interference of multipath components.

#### 3.3.2. Retail Store Measurement

To approach the realistic scenario of common measurements in a retail store, they are performed in a proof-of-concept manner by using an Impinj speedway RFID commercial reader (Impinj, Inc., Seattle, WA, USA) with circularly polarized antenna of 8 dBi gain and the transmitting power of 26 dBm in European RFID frequency band for all investigated scenarios. As the commercial reader has limited measurement data analysis, we also used a sniffer tool in order to recognize the messages of reader and evaluate the communication between a commercial RFID reader and a tag [25]. It is a tool developed in the GNU radio open-source platform that uses a Universal software radio peripheral (USRP) hardware for recording the messages from a reader during tag reading. Originally, the tool was built for larger bandwidth USRP2 measurement platform for US frequency band (902–928 MHz). However, due to the available equipment, and for purposes of given measurements, the software was tuned to use USRP1 hardware which was enough, as EU frequency band uses more narrow frequency range (865–868 MHz). The sniffer tool can demodulate/decode reader commands according to the EPC Gen2 RFID protocol [1]. According to the given standard, at first, the reader broadcasts a QUERY command, which sets the size of the frame, which consists of time slots wherein the tags are taking a random time slot within the given frame. Therefore, each frame is interrogated in a slot-by-slot fashion, where the next slot is interrogated after a QREP command. Once the slot, which contains the tag, replies to the reader, the reader acknowledges tag response (with ACK command) and the sniffer tool is able to understand if there was a tag reply within the frame, i.e., between two consecutive frames.

Figure 14 shows the measurement setup and the positions of sniffer and Impinj antennas in a retail store. All measurements are performed with a tag moved along the trajectory around the shelves (as in Figure 12). Ten laps are performed during each measurement and the sniffer saved just one log file per measurement. When the Impinj reads the tag attached to a cardboard or some product always in the same orientation to the reader, the sniffer collects the messages from their communication (QUERY, QREP, ACK, ERROR, etc.) and writes them into the log file (the reading process is explained in [26]). If there is a large number of tag responses, then the log file is large, but for the tag positions where it does not respond, the log file is small (fewer lines, i.e., messages). Therefore, the measurement log file is analyzed by taking the number of ACKs (this is the acknowledgement message from the sniffer received only when the tag is read) per one second of total measurement. Consequently, the number of ACKs, $N_{ACK}$ is bigger in LOS area where Impinj is reading the tag well, but in non-LOS area; this number is small or zero.

The measurement results retrieved from a commercial reader Impinj are given in Table 5 and Table 6. The average number of reads per second is obtained as the total number of reads divided by the total duration of the performed measurement. From Table 5 where Higgs 2, 3, and 4 are attached to a cardboard, we can see that the best reading results are obtained for Higgs 4 and the worse for Higgs 2. When the Higgs 4 tag is attached to a retail product, its response is decreased and depends on a product material and its size. The lowest read rate is obtained for a tag attached to a large water canister and the highest for plastic products. This agrees with the conclusions from Section 2. It must be noted that Higgs 4 attached to metallic products did not show any response.

Figure 15 and Figure 16 depict the results retrieved from sniffer’s log file, number of ACKs vs time. Generally, one can notice that there are drops or zeros of responses (in non-LOS) after some periods of high numbers of readings when the tag is in the LOS area. This agrees with the simulated measurement scenario (Figure 13). Again, Higgs 2 shows the worst performance regarding the number of responses as well as their variation along the measurement trajectory, in relation to both Higgs 3 and Higgs 4. When Higgs 4 is attached to a retail product, the number of responses decreases, and the variation of its responses increases due to multipath propagation and material effects discussed earlier. If the product is larger in size, then the deterioration of tag antenna properties is more significant (see Figure 3 and Table 3) which causes degraded response with large variations in the number of responses along the trajectory (lower graph in Figure 16). The responses of a tag attached to low-loss dielectric material are given in Figure 17 for a pack of plastic plates and trash bags. Both products are made of plastic and show an equal number of responses per second (Table 6). The results show a similar pattern of $N_{ACK}$ along a trajectory but still different response drops are noticed caused by small differences in multipath channel and influence of products on tag antenna properties due to their different size and shape.

To sum up, the measurement results showed the highest response of Higgs 4 and when attached to product, the degraded multipath pattern with lower response for water products in relation to plastics.

## 4. Summary and Comparison of Simulation and Measurement Results

The fundamental aspects of passive RFID system (from Equations (Equation 2) to (Equation 4)) show that in free space, without any obstacles and products, there are limitations due to chip and antenna design, frequency, power level, and orientation of tag to reader. In this empirical study, the RFID system is placed into a realistic retail store environment: a realistic simulation model and a real store with tag responsiveness measurements.

First, the simulations of well-known effects related to the variation of antenna radiation characteristics when placed on some material are given. These simulations are performed in the context of a retail store scenario to visualize and associate EM phenomena with the measurement results. The tag is attached to products (modeled as a cylinder) of different electrical properties: water (representing all liquids), metal (representing all metallic products) and plastic (low-loss dielectric properties taken for all non-metallic and non-liquid products). The radiation properties of tag antenna attached to a retail product are degraded significantly depending on their size and material properties. For all three materials, the impedance match between the antenna and chip is degraded, the tag antenna directivity is changed and radiation efficiency decreased (Figure 2 and Table 1). This effect is shown and agrees well with the results of retail store measurements when the tag is placed on cardboard, water products (bottle of water and water canister), and plastic products (a pack of plastic plates and trash bags). The measurements are performed in a concept-proof manner with commercial Impinj reader and sniffer tool (using USRP) in order to evaluate messages from the reader during the tag trajectory around the shelves. The number of tag responses per second was higher for the tag attached to cardboard than for the cases of water and plastic products. The passive Higgs 4 tag did not respond to metal products. Water product has more significant influence than the plastics, especially when the canister of water was used (Figure 16). This is due to its larger size, which causes greater degradation of tag radiation properties, as shown in simulations (Figure 3 and Table 3).

The variation of power impinging on the tag caused by multipath phenomena in a retail store was also evaluated by simulations and measurements. A realistic simulation model of a retail store with concrete walls, ceiling and floor, metallic shelves and refrigerators is used for various scenarios of receiver setups and product arrangements. The results for evaluation of multipath environment are given as power distribution in the store (horizontal and vertical receiver grid), as CDFs of all scenarios related to products’ materials on the shelves, receiver power along the trajectory and number of reads during the measurement trajectory. For the system setup in our work, the probability that Higgs 2 will not be read ($P_{tag}$ < $P_{tag,sens}$) for standalone tag at random position in the store (2D receiver grid) is 49% on average for scenarios of empty shelves, water, metal, and plastic products on the shelves (with the worse results for metal products shown in Figure 11 and Table 4). Higgs 4 shows better performance regarding power distribution and 17% lower probability of not being read (in average for mentioned cases). The worst case of all scenarios is the most realistic one, and that is a tag attached to a product (water product) and moved around the store (or along the trajectory) with mixed products on the shelves. This is due to lower antenna gain caused by the losses of a product material. It shows very low probability of being read (13% for Higgs 2 and 19% for Higgs 4) and significantly decreased range in power distribution results.

Both simulation and measurement results agree and show that due to multipath propagation, unexpected power level drops can happen in a LOS area and very near the reader antenna, as well as some receiver positions with unpredictable higher power level in non-LOS area far from reader (Figure 13, Figure 15, Figure 16 and Figure 17). The number of reads per second obtained from measurements also show significant degradation of the multipath pattern along the trajectory and a lower number of responses when the tag is attached to a product (Figure 16 and Figure 17). This means that the effect of product material on tag performance can have stronger influence on retail store RFID application than the multipath phenomena, depending on material used. In that sense, we can provide a rough measure of the EM effects influencing the application of RFID in retail store. Specifically, in Figure 13, if we consider and compare the free space results—tag sensitivity level measured in anechoic chamber (for measurements it is the closest to free space) and tag moving along the trajectory in free space—with the realistic channel results—standalone tag and tag on water product moving along the trajectory in retail store channel—then we can observe the following:The multipath channel influence:In comparison with a tag in free space, the multipath channel causes drops of received power by roughly 10 dB, as well as power gains by 10 dB, between close receiver positions in a LOS area (e.g., receiver position 6). These oscillations are more emphasized in non-LOS areas and a difference in chip sensitivity can help improve the multipath susceptibility (in our case the difference is 4.2 dB but in the rest of the frequency band it can be more, Figure 6). In the product arrangement of the store, the improvement of multipath power gain can be achieved by placing the metallic products on the shelves (Figure 11 and Table 4). The product material influence:When the tag is attached to a water product (in our case), the received power is lower by 14 dB (due to 14 dB lower antenna gain) for all receiver positions, so tags generally have no or low power to activate, but multipath propagation improves the coverage because of multipath power gain at some receiver points. This product influence can be less or more emphasized depending on different materials, their shape, and size. In Figure 16 and Figure 17 tags attached to plastic materials show better multipath susceptibility (less signal variation over time) and a higher number of responses than tags on water products, i.e., the average number of reads per second for plastics is 35 whereas for a bottle of water it is 10 and for a water canister only 4 (the highest one is 46 for tag on cardboard, as expected). It is important to note that the multipath propagation paths can be different when the directivity of the tag antenna is changed due to material effect (Figure 2 and Figure 3).

In this study, a relatively low probability of reaching tag activation threshold is obtained in practically all scenarios (in the best case 0.75 for metal objects on the shelves, Figure 11 and Table 4) mostly due to the chosen transmitter position covering a small LOS area. The optimum signal coverage in the store and reaching the highest performance of the system is beyond the scope of this paper. Since the estimation of the RFID application in a retail store is practically a compromise between the product and multipath channel influence on tag responsiveness, there is an implication that different store layouts, reader positions, and product arrangements could improve the system performance, e.g., to place water products closer to the reader than plastic products, to place metallic products in strategic places because this improves the multipath power gain, or cover non-LOS areas with additional reader antennas. However, this requires detailed further empirical analysis on the subject in our future work.

## 5. Concluding Remarks

The potential of IoT application in a retail store is very high but, in the physical layer, available channels and propagation resources limit the IoT possibilities. From the EM point of view, the retail store propagation channel, as well as the electromagnetic compatibility of a tag antenna with retail products, needs to be thoroughly investigated.

In short, the simulations of well-known effects are performed in the context of a retail store in order to visualize, understand, and associate EM phenomena with measurements in retail store. The analysis and the estimation of the compromise between EM phenomena influencing the tag responses can help to build and rearrange store layouts for IoT use.

In future work, the focus will be on the extension of the empirical study given in this paper. The plan is to measure and simulate different setups in the store related to product arrangement, reader antenna positions, and store layout. The goal is to provide a methodology for building an IoT retail store from an EM point of view.

## Figures and Tables

**Figure 1 sensors-20-02038-f001:**
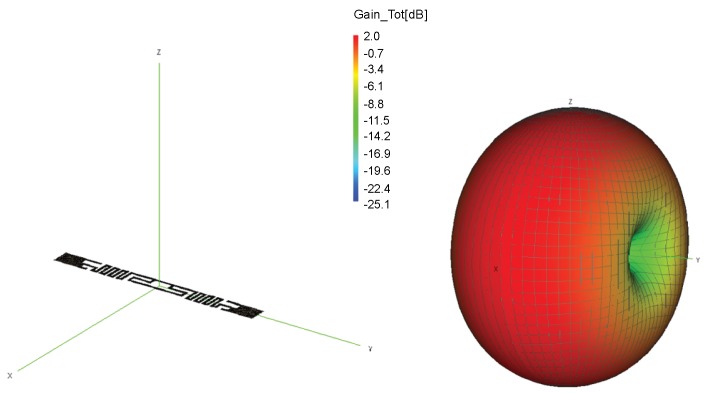
The tag structure and 3D radiation pattern at 920 MHz in free space.

**Figure 2 sensors-20-02038-f002:**
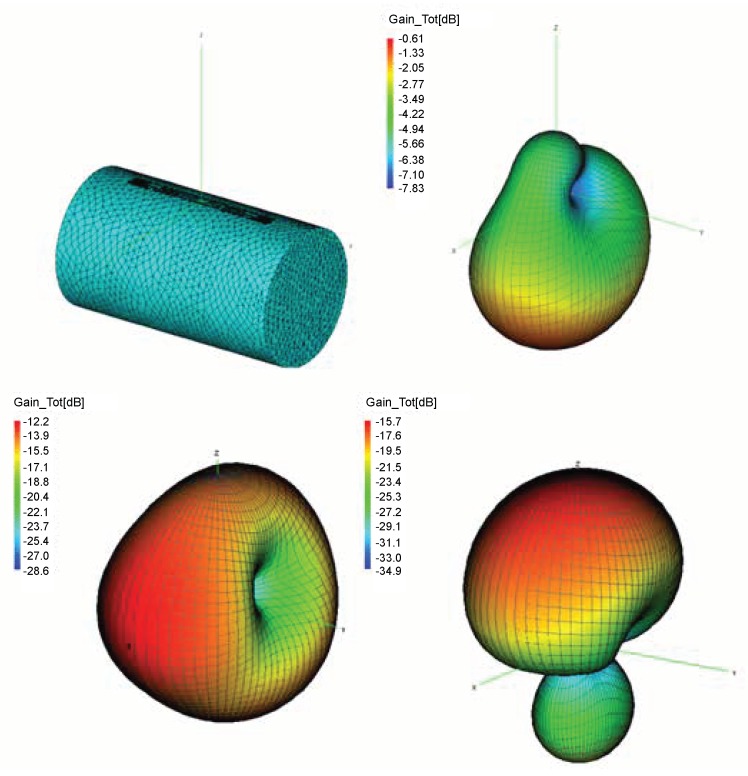
The structure of tag attached to a cylinder and 3D radiation pattern at 920 MHz of tag attached to the cylinder (**upper left**) made of plastic (**upper right**), water (**lower left**) and metal (**lower right**).

**Figure 3 sensors-20-02038-f003:**
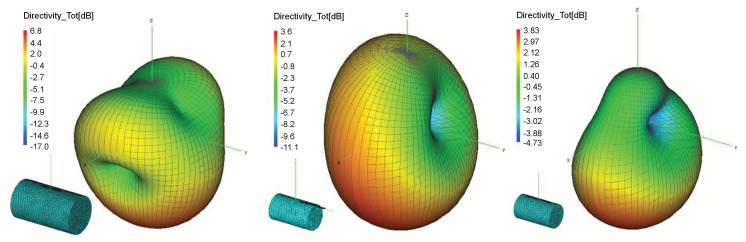
3D directivity patterns at 920 MHz for tag attached to the twice larger plastic cylinder (**left**), with minor part in the air (**center**), and at 3-mm distance from the plastic cylinder (**right**).

**Figure 4 sensors-20-02038-f004:**
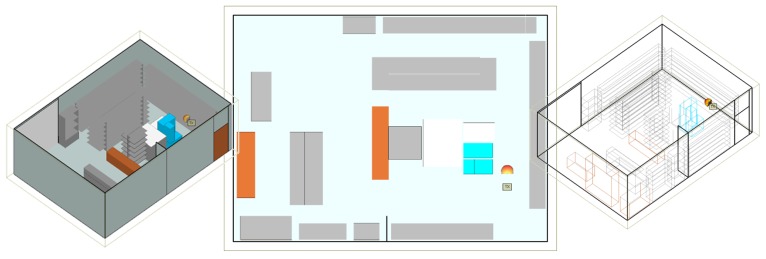
Wireless Insite simulation model of a real retail store (different views).

**Figure 5 sensors-20-02038-f005:**
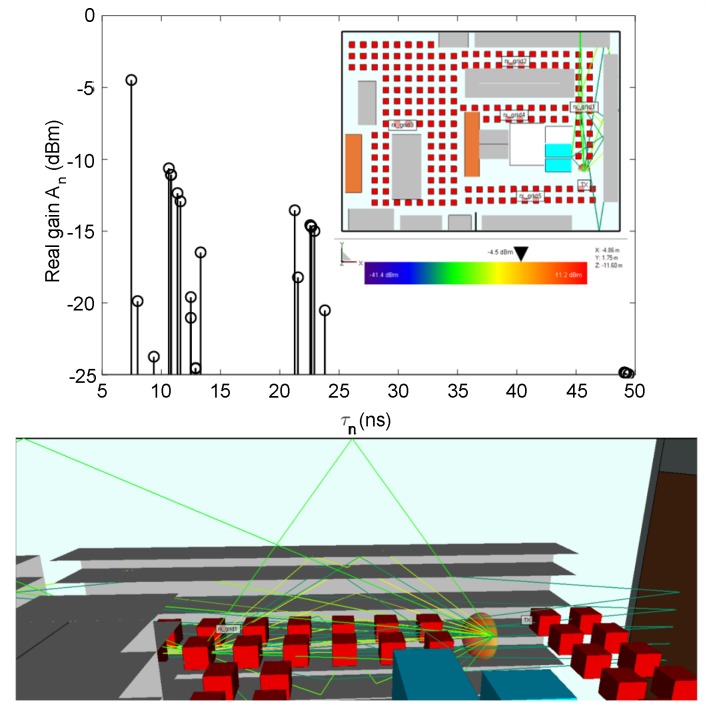
Impulse response at the receiver point 11 in receiver grid 1 (LOS area) with 2D and 3D view of the propagation paths in empty shelves scenario.

**Figure 6 sensors-20-02038-f006:**
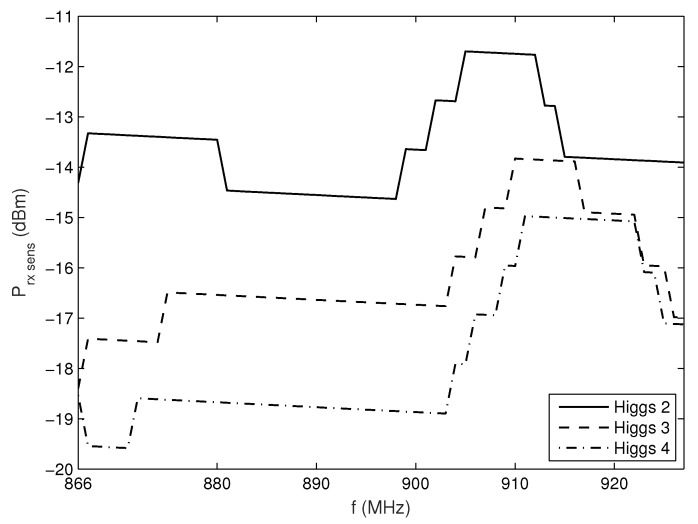
Measured sensitivity levels for Higgs 2, Higgs 3, and Higgs 4.

**Figure 7 sensors-20-02038-f007:**
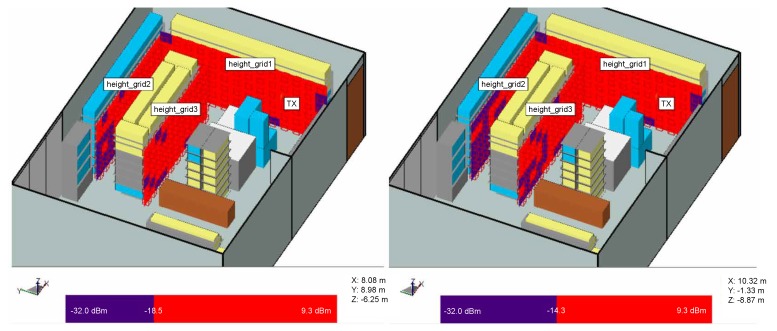
Power distribution for simulation model with mixed products on the shelves and vertical receiver grid, Higgs 2 (**right**) and Higgs 4 (**left**).

**Figure 8 sensors-20-02038-f008:**
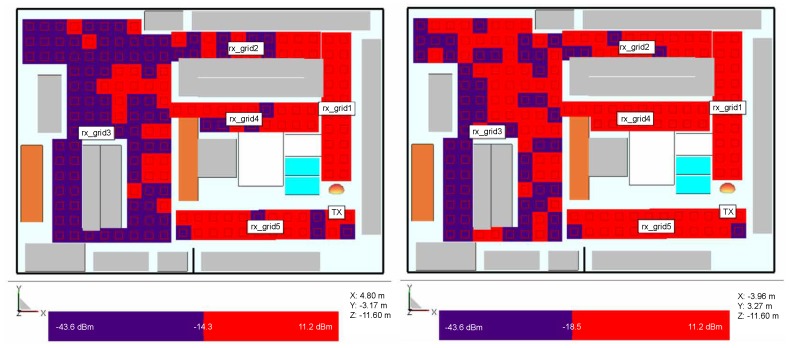
Power distribution for simulation model with empty shelves and standalone tag, Higgs 2 (**left**) and Higgs 4 (**right**).

**Figure 9 sensors-20-02038-f009:**
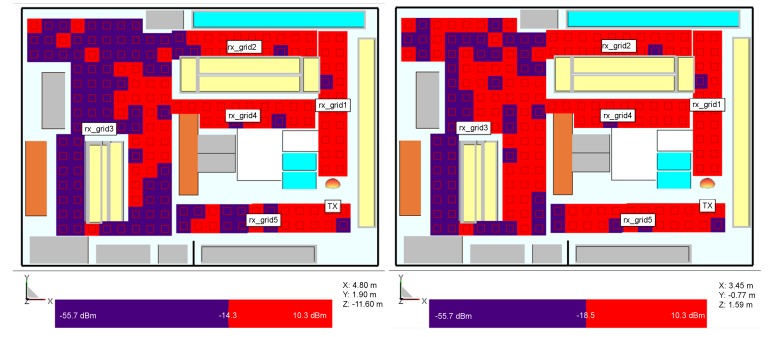
Power distribution for simulation model with mixed products and standalone tag, Higgs 2 (**left**) and Higgs 4 (**right**).

**Figure 10 sensors-20-02038-f010:**
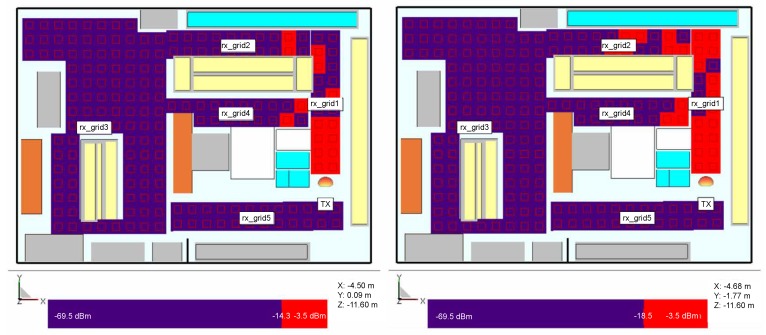
Power distribution for simulation model with mixed products with tags on water product, Higgs 2 (**left**) and Higgs 4 (**right**).

**Figure 11 sensors-20-02038-f011:**
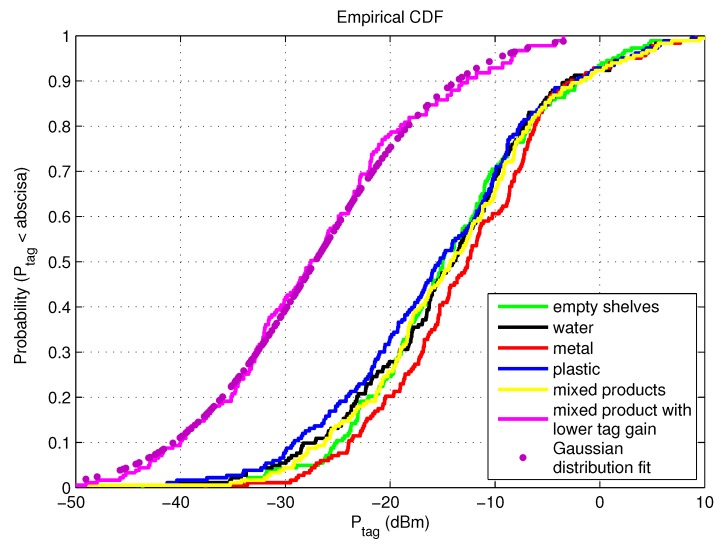
Empirical CDFs for all simulated scenarios of 2D receiver grid.

**Figure 12 sensors-20-02038-f012:**
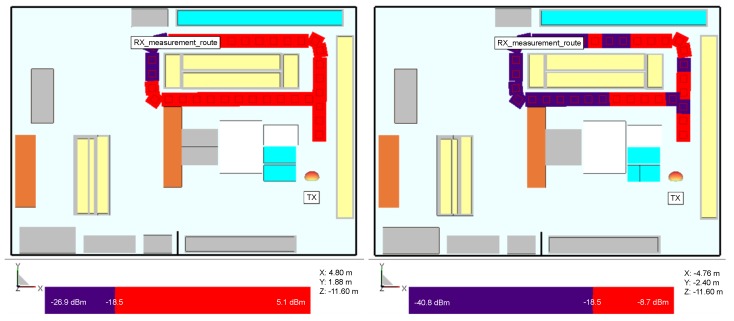
Simulation results of $P_{tag}$ distribution along a trajectory of a Higgs 4 tag attached to a cardboard (**left**) and water product (**right**).

**Figure 13 sensors-20-02038-f013:**
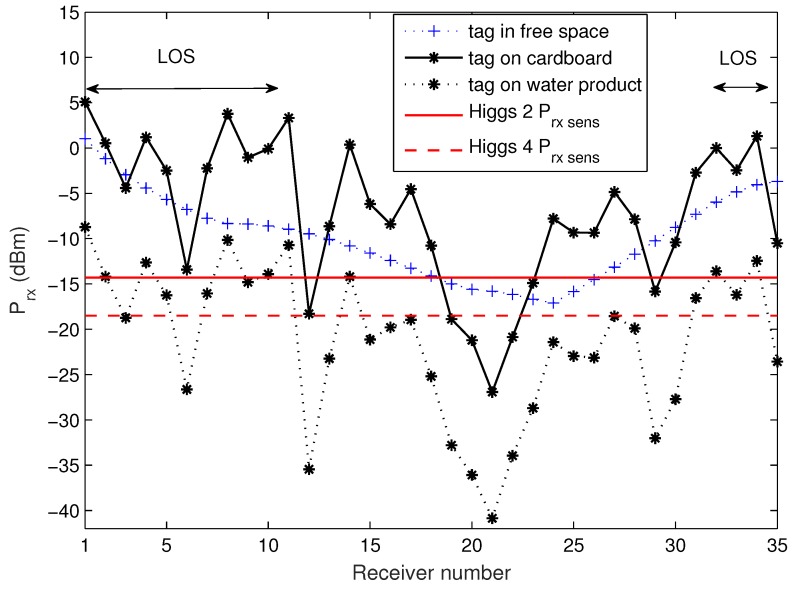
Simulation results of $P_{tag}$ variation along the receiver trajectory.

**Figure 14 sensors-20-02038-f014:**
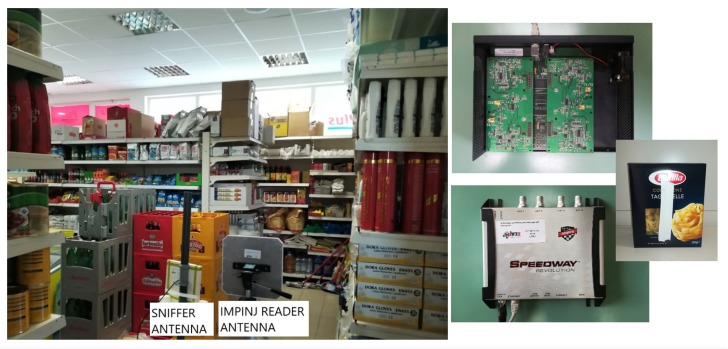
A photo of a measurement setup in a retail store, Impinj reader, SDR, and tag on a product.

**Figure 15 sensors-20-02038-f015:**
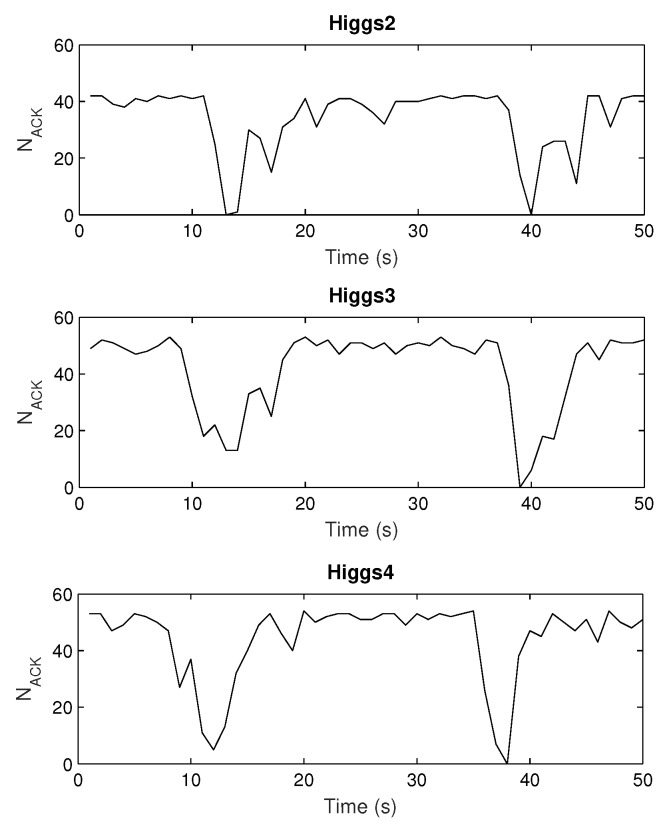
Sniffer measurements of tag responses when attached on cardboard.

**Figure 16 sensors-20-02038-f016:**
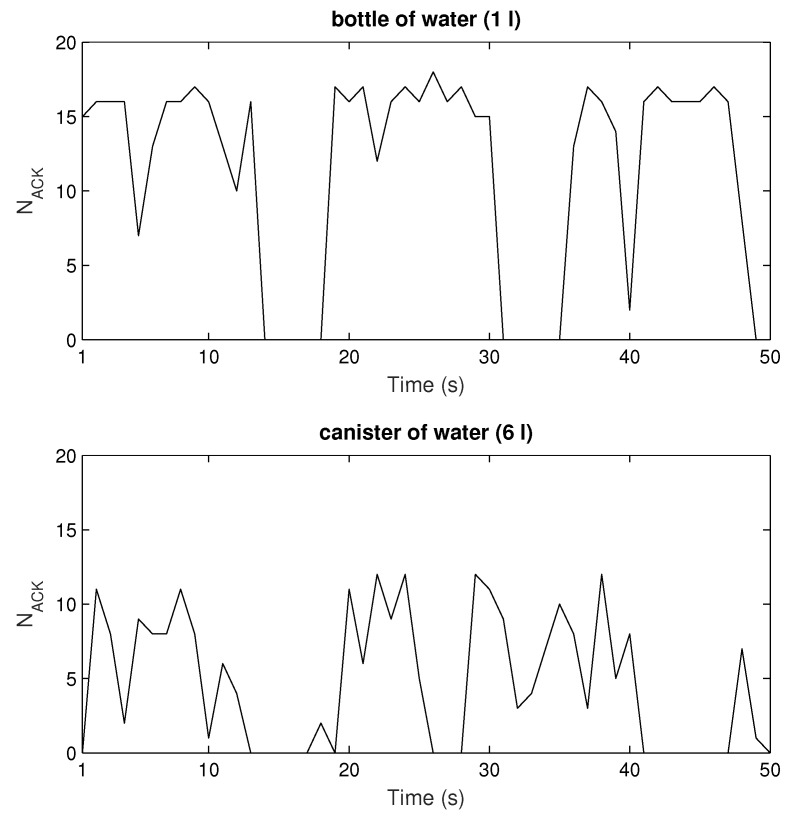
Sniffer measurements of tag responses when attached on water products of different size.

**Figure 17 sensors-20-02038-f017:**
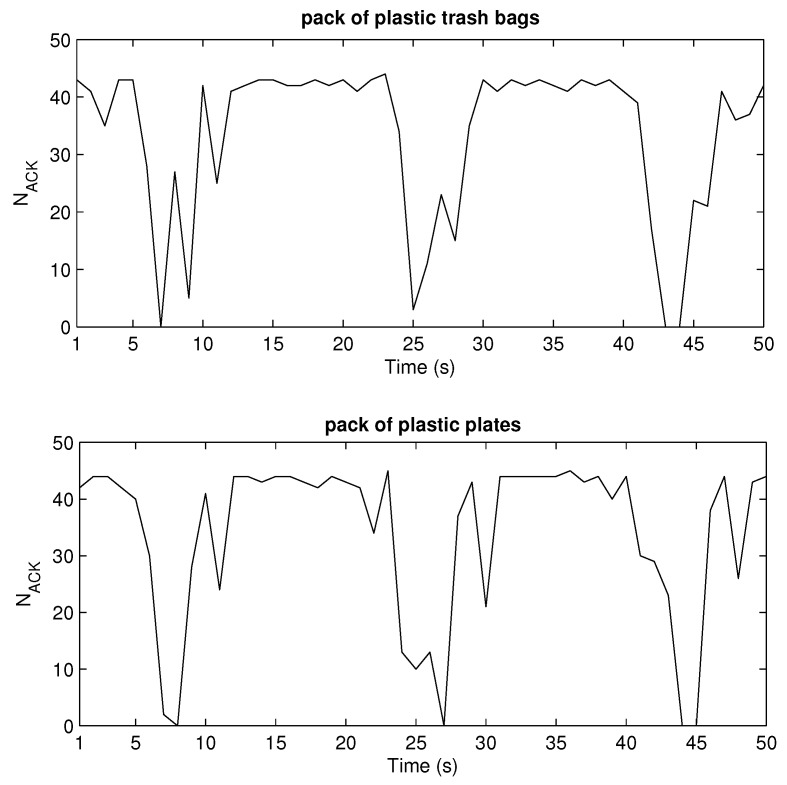
Sniffer measurements of tag responses when attached on plastic products of similar properties.

**Table 1 sensors-20-02038-t001:** Impedance and radiation efficiency results of tag antenna for different scenarios at 920 MHz.

Scenario	Z (Ω)	$e_{r}$ (%)
free space	51.0 + j237.7	97.8
water	7.9 + j198.9	5.94
plastic	3.4 + j185.2	55.4
metal	0.3 + j169.1	0.57

**Table 2 sensors-20-02038-t002:** Impedance, radiation efficiency and gain results of tag antenna attached to plastic product in different scenarios at 920 MHz.

Scenario	Z (Ω)	$e_{r}$ (%)	Gain (dBi)
d = 3 mm	12.5 + j172.8	75.8	2.63
d = 6 mm	32.4 + j161.6	81.0	3.98
Minor antenna part in air	7.1 + j155.0	49.9	0.60
Major antenna part in air	12.2 + j178.1	80.0	2.30
Tag attached (d = 0.1 mm) to larger cylinder	3.2 + j200.3	42.9	3.1

**Table 3 sensors-20-02038-t003:** Material electrical properties of a retail store simulation model.

Material	$\epsilon_{r}$	σ (S/m)
wall concrete	7.00	0.015
floor and ceiling concrete	15.0	0.015
wood	5.00	0.000
glass	2.40	0.000
water	81.0	0.220
plastic	3.50	0.002

**Table 4 sensors-20-02038-t004:** Statistical parameters of $P_{tag}$ at each receiver position in a 2D receiver grid for all examined scenarios.

	AVG	STD	Probability ($P_{\mathrm{tag}}$ < $P_{\mathrm{tag},\mathrm{sens}}$)	
Material of Products on the Shelves	$P_{\mathrm{tag}}$ (dBm)	$P_{\mathrm{tag}}$ (dB)	Higgs 2	Higgs 4
Empty shelves	−14.3	9.1	0.52	0.33
Water	−14.5	9.7	0.49	0.31
Metal	−12.7	8.6	0.44	0.25
Plastic	−15.4	10.3	0.52	0.38
Mixed (water, metal, plastic)	−14.2	9.7	0.49	0.32
Mixed (water, metal, plastic)				
with lower tag antenna gain	−27.2	10.5	0.87	0.81

**Table 5 sensors-20-02038-t005:** Impinj reading results along the measurement trajectory of a tag attached to a cardboard.

Tag	Average Number of Reads Per Second
Higgs 2	35
Higgs 3	43
Higgs 4	46

**Table 6 sensors-20-02038-t006:** Impinj reading results along the measurement trajectory of a Higgs 4 tag attached to different products.

Tagged Product	Average Number of Reads Per Second
bottle of water 1 l	10
canister of water 6 l	4
pack of plastic plates	35
pack of plastic trash bags	35

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
