# Peer review of "Analysis of Passive RFID Applicability in a Retail Store: What Can We Expect?"

_sensors, 2020, doi:10.3390/s20072038_

Round 1
Reviewer 1 Report
The paper provides both simulation and experiment/measurement results to investigate the applicability of passive RFID in a retail store.
The authors have devoted a lot of effort in doing the simulations and experiments, and I agree that these thorough measurement results are valuable not only for practical implementation of RFID systems but also for the research community. In that sense, this work does have research contribution.
A few comments on how to improve the paper are as follows:
First of all, it would be nice if the findings of this work can be categorized into 'expected' and 'counter-intuitive'.
For example, it is expected that the propagation characteristics will differ depending on the material on which the RFID tag is attached to. The only question is "by how much?". Also, it is expected that LOS/NLOS will have significant differences, position of the reader will have significant impact, and multipath effect will influence performance.
Of course, verifying these conjectures/intuitions through real measurements are important; I totally agree and respect that. However, verifying/confirming and getting numbers for somewhat expected behavior is one thing, and discovering counter-intuitive behavior is another. So, It would be nice if these two are separated, and clearly emphasized.
On a similar line, it is unclear how well the simulation results agree with the experiment/measurement results.
Also, for each of the simulation and experiment, it would be nice if the 'purpose' of that sim/exp is clearly stated at the beginning, and the 'key finding' is emphasized at the end of each (sub)section. The authors did a lot of experiments, and it is easy to get lost on why it is done and what it meant to show.
Section 4 was a good summary of what has been done, but I think it is better to separate the 'summary of what has been done' and 'conclusion of findings'.
More fundamental question is this: who would use a (RFID) system that can only read 10~50% of the messages? This is useless system; Real industrial systems require 99% reliability, and no company/retailer will use a system that has 10~50% reliability.
(for example, a system that wanted 99%, not 95% ==> https://doi.org/10.1109/ACCESS.2019.2950886)
When we conduct experiments, we need to design the scenario and setup. Depending on how you design the scenario and setup (e.g. distance, tx power, physical environment), you can get totally different results even with identical devices.
The experiments that the authors have done are valuable. However, IMHO, the authors should have suggested a setup at which 90+% was possible (For example, with more number of readers?), and show measurement results in that setup as well. That way, the finding would be "if you want to use RFID system in a retail store, you need to consider these things. This setup will not work, and that setup will work... you need more readers, or current generation RFID is junk, or you must deploy RFID this way" and so on.
To summarize, this paper presented measurement results of real RFID tags, showing that the performance is highly variable, sometimes very low, and thus it is challenging to use in a real retail store. The paper should have also suggested how it could be made better (and get 99% so that it can be used), or argue that RFID is currently not good enough for real deployment in retail stores. --> this conclusion is missing.
Thanks
Reviewer 2 Report
The authors approach a subject of great importance in IoT, the basic support layer, that is in most cases RFID, due to costs and convenience. The stated purpose of the paper is to perform an empirical analysis on the limitations of RFID technology in a retail scenario. EM analysis is performed using simulations and measurements when tags are attached to different materials.
The results support the conclusions, the experiments are thoroughly described and the paper has value. However there are some issues which should be fixed before an eventual publication.
-figures and tables could be better edited, especially from page 15 onwards. Resolution is poor and tables could be better formatted.
-bibliography is scarce and outdated. In fact paper is lacking a proper state-of-art chapter. This should be fixed.
Reviewer 3 Report
The interest of the topic has decrease with the time (the authors themselves talk about 12 years of none research on RFID, which is not completely right). If nobody did anthing on the topic during the last 12 years, this means that the interest is relatively low nowadays.
The main lack I found is a deeper comparison between simulation and measurement results within the shop. Perhaps a section devoted of that comparison would be a good solution.
Regarding the measurements, the main problem is that they are performed using non-calibrating equipment. When you use lab equipment, you have calibration sheets and then you can predict an uncertainty budget, which provides a range of validity to the provided data. This is not possible in the shown experiments, so that we are not having scientific measurements. What you present is a concept proof: this is interesting engineeringly speaking, but they are not strict scientific results. I think that this must be stated along the manuscript: you cannot provide uncertainty data for your measurements, so you a proof of concept, not scientific measurements.
I also suggest to reduce the sections regarding the EM development: there are many well known situations and effects, figure 3 can be better explained by a table, and so on.
Regarding the introduction, and hte comment on the RFID work in the last 12 years, I suggest to read the following papers:
* Suijing, H. Han, L. Wenling, and Z. Zhongyuan, “The Design and Implementation of the Warehouse System Based on RFID and Mobile Devices,” 2nd International Conference on Computer and Automation Engineering, Singapore, Singapore,
February, 2010
* M. Trebar, A. Grah, A. A. Melcon, and A. Parreno, “Towards RFID Traceability Systems of Farmed Fish Supply Chain,” 19th International Conference on Software, Tele communications and Computer Networks, Hvar, Croatia, September, 2011
* Y. Zhang, K. Yemelyanov, X. Li, and M. G. Amin, “Effect of metallic objects and liquid supplies on RFID links,” in Proceedings of the IEEE International Symposium on Antennas and Propagation and USNC/URSI National Radio Science Meeting
(APSURSI ’09), Charleston, SC, USA, June 2009
* Iñigo Cuiñas, Robert Newman, Mira Trebar, Luca Catarinucci, Alejandro A. Melcon, "RFID-based traceability along the food production chain", IEEE Antennas and Propagation Magazine, vo.56, no.2, pp.196-207, April 2014
Round 2
Reviewer 1 Report
The authors have addressed most of my comments to some extent.
Although this reviewer really wanted to know how to (or whether it is possible to) improve the performance of a RFID system to a practically usable level (~95%+) in a real retail store setting (by some system level configuration or something), I do understand that that might be another level of work which could be beyond the scope of this work.
Other than that, I do not have any further comments.
Thanks.
Reviewer 3 Report
After reviewing the new version of the manuscript, I think authors have solved all my concerns about this paper, and so that I recommend to accept the manuscript as is.